# The Relationship Between Smartphone and Game Addiction, Leisure Time Management, and the Enjoyment of Physical Activity: A Comparison of Regression Analysis and Machine Learning Models

**DOI:** 10.3390/healthcare13151805

**Published:** 2025-07-25

**Authors:** Sevinç Namlı, Bekir Çar, Ahmet Kurtoğlu, Eda Yılmaz, Gönül Tekkurşun Demir, Burcu Güvendi, Batuhan Batu, Monira I. Aldhahi

**Affiliations:** 1Department of Physical Education and Sports Teaching, Faculty of Sport Science, Erzurum Technical University, 25050 Erzurum, Türkiye; sevinc.namli@erzurum.edu.tr (S.N.); eda.yilmaz@erzurum.edu.tr (E.Y.); 2Department of Physical Education and Sports Teaching, Faculty of Sport Science, Bandırma Onyedi Eylül University, 10200 Balıkesir, Türkiye; carbekir@gmail.com; 3Department of Coaching Education, Faculty of Sport Science, Faculty of Sport Science, Bandırma Onyedi Eylül University, 10200 Balıkesir, Türkiye; kurtogluahmet18@gmail.com; 4Independent Researcher, Abu Dhabi 20029, United Arab Emirates; gonultekkursun@hotmail.com; 5Department of Coaching Education, Faculty of Sport Science, Yalova University, 77100 Yalova, Türkiye; burcu.guvendi@yalova.edu.tr; 6Department of Management and Organisation, Kağızman Vocational School, Kafkas University, 36000 Kars, Türkiye; batuhan.batu@kafkas.edu.tr; 7Department of Rehabilitation Sciences, College of Health and Rehabilitation Sciences, Princess Nourah bint Abdulrahman University, P.O. Box 84428, Riyadh 11671, Saudi Arabia

**Keywords:** smartphone addiction, game addiction, leisure time management, physical activity, machine learning, artificial intelligence

## Abstract

**Background/Objectives:** Smartphone addiction (SA) and gaming addiction (GA) have become risk factors for individuals of all ages in recent years. Especially during adolescence, it has become very difficult for parents to control this situation. Physical activity and the effective use of free time are the most important factors in eliminating such addictions. This study aimed to test a new machine learning method by combining routine regression analysis with the gradient-boosting machine (GBM) and random forest (RF) methods to analyze the relationship between SA and GA with leisure time management (LTM) and the enjoyment of physical activity (EPA) among adolescents. **Methods:** This study presents the results obtained using our developed GBM + RF hybrid model, which incorporates LTM and EPA scores as inputs for predicting SA and GA, following the preprocessing of data collected from 1107 high school students aged 15–19 years. The results were compared with those obtained using routine regression results and the lasso, ElasticNet, RF, GBM, AdaBoost, bagging, support vector regression (SVR), K-nearest neighbors (KNN), multi-layer perceptron (MLP), and light gradient-boosting machine (LightGBM) models. In the GBM + RF model, probability scores obtained from GBM were used as input to RF to produce final predictions. The performance of the models was evaluated using the R^2^, mean absolute error (MAE), and mean squared error (MSE) metrics. **Results:** Classical regression analyses revealed a significant negative relationship between SA scores and both LTM and EPA scores. Specifically, as LTM and EPA scores increased, SA scores decreased significantly. In contrast, GA scores showed a significant negative relationship only with LTM scores, whereas EPA was not a significant determinant of GA. In contrast to the relatively low explanatory power of classical regression models, ML algorithms have demonstrated significantly higher prediction accuracy. The best performance for SA prediction was achieved using the Hybrid GBM + RF model (MAE = 0.095, MSE = 0.010, R^2^ = 0.9299), whereas the SVR model showed the weakest performance (MAE = 0.310, MSE = 0.096, R^2^ = 0.8615). Similarly, the Hybrid GBM + RF model also showed the highest performance for GA prediction (MAE = 0.090, MSE = 0.014, R^2^ = 0.9699). **Conclusions:** These findings demonstrate that classical regression analyses have limited explanatory power in capturing complex relationships between variables, whereas ML algorithms, particularly our GBM + RF hybrid model, offer more robust and accurate modeling capabilities for multifactorial cognitive and performance-related predictions.

## 1. Introduction

The integration of digital technologies into daily life has profoundly transformed the way individuals structure and engage in leisure activities, influencing both the nature and accessibility of recreational experiences. In particular, smartphones and digital games have taken an important place in daily life, as they have a socialization function and facilitate access to information [1]. Excessive engagement with digital technologies has been shown to negatively affect leisure time management [2], minimize participation in physical activities [3], and cause physical and mental health problems in the long term [4]. Smartphones and digital games, which are indispensable building blocks of the digital age [5], put pressure on leisure time and cause physical effects on individuals. Furthermore, the increasing reliance on digital devices as the primary source of entertainment and engagement plays a crucial role in shaping leisure behaviors [6]. While digital platforms may provide a temporary escape from boredom; over time, this reliance can contribute to physical inactivity and potential addiction [7]. These findings highlight the need for balanced leisure activities that integrate both digital engagement and physical participation to promote overall well-being.

It is observed that smartphone (SA) and game addiction (GA) create changes in individuals’ planning for leisure time management [8] and may lead to a decrease in the time allocated to physical activity due to a passively screen-dependent life [9]. From a motivational perspective, prolonged engagement with digital devices can also diminish the happiness derived from physical activity [10]. Conversely, individuals who prioritize productive leisure time and actively participate in physical activities are generally less prone to SA and GA engagement. Studies suggest that individuals who engage in regular physical activity experience lower stress levels, stronger social connections, and an improved overall quality of life. Effective leisure time management has been associated with psychological and physical benefits, similar to those observed in individuals who engage in regular physical activity. A recent study suggested that university students who manage their leisure time effectively exhibit a lower risk of smartphone addiction (SA) [11]. However, recent studies have shown that SA and GA are influenced by many environmental factors, and different methods are used to diagnose and identify such addictions.

ML and AI have been widely used in recent years for the diagnosis and identification of behavioral disorders, such as SA and AD. ML models, in particular, allow the risk factors of such addictions to be analyzed with high accuracy. In the study conducted by Lee and Kim, a model was created regarding the SA levels of participants according to age, job, and gender. Different algorithms were created for this model, and the highest accuracy rate was achieved with the random forest (RF) algorithm [12]. Giraldo-Jiménez et al. predicted SA with a questionnaire on musculoskeletal symptoms and risk factors and achieved the highest accuracy rate with the support vector machine (SVM) algorithm [13].

In the existing literature, studies linking SA and GA to psychosocial variables are mostly limited to separate regression or single ML algorithms; research simultaneously addressing LTM and EPA and comparing their predictive power with a hybrid model is rare. In this study, a two-stage GBM + RF hybrid model using LTM and EPA as inputs was developed, and its performance was compared with classical regression methods such as lasso, ElasticNet, RF, GBM, AdaBoost, bagging, SVR, KNN, MLP, and LightGBM. Our hypotheses are as follows: (H1) increasing LTM scores will significantly reduce both SA and GA, (H2) high EPA scores will have a reducing effect on SA and GA, and (H3) GBM + RF will offer higher generalizability and prediction accuracy compared to classical methods and single ML algorithms. This approach aims to demonstrate the superiority of hybrid models for multidimensional behavior prediction.

## 2. Materials and Methods

### 2.1. Study Design and Participants

In this study, the survey model, one of the quantitative research methods was used. Survey studies are conducted to describe and analyze a current situation [14]. This statistical method was used to examine the relationships between variables such as SA and GA, LTM, and EPA.

This study included high school students aged 15–19 years who were actively engaged in educational activities during the 2023–2024 academic year. A non-probability convenience sampling method was employed. To maintain the integrity of the study, individuals with continuous absenteeism, diagnosed mental health conditions, inclusion in mainstreaming programs, or acute illnesses that impaired their ability to comprehend the scale’s questions were excluded. A power analysis was conducted to determine the required sample size using the following parameters: α = 0.05, power (1 − β) = 0.80, odds ratio = 0.7, and Pr(Y = 1|X = 1) HO = 0.045. The results indicated that a minimum of 1087 participants was needed to achieve an actual power of 0.80. Based on this analysis, a total of 1107 students (416 male, 691 female) were included in the final sample.

This study was conducted in accordance with the principles of the Declaration of Helsinki and the ethical guidelines. During the data collection process, Google Forms were used to collect online responses to scales related to smartphone and game addiction, leisure time management, and the enjoyment of physical activity. Participants were informed that participation in the study was voluntary, and confidentiality principles were followed. In order to collect the data, the relevant institutions and individuals were contacted via email, and informed consent was obtained from the participants. In addition, the study was initiated after the necessary informed consent was obtained from the parents or legal representatives of the participants. After a 45-day data collection period, statistical analyses were performed on the completed and correctly completed questionnaires. The study was approved by the ethics committee of the Bandırma Onyedi Eylu University Social and Human Sciences Ethics Committee on 14/04/2023 and numbered 2023-4.

### 2.2. Data Collection Tools

For this study, scales related to SA, GA, LTM, and EPA, and a personal information form created by the researchers, were used.

#### 2.2.1. Smartphone Addiction Scale

The Smartphone Addiction Scale, developed by Şata and Karip, was used to determine the level of smartphone addiction. The scale is a valid and reliable tool for measuring individuals’ smartphone usage habits and addiction levels. This scale, which consists of a single dimension and contains 10 items, evaluates the impact of smartphone use on individuals’ daily lives. The Cronbach’s alpha (α) reliability coefficient of the scale was 0.96, indicating that the scale is highly reliable [15].

#### 2.2.2. Leisure Time Management Scale

The Leisure Time Orientation Scale, developed by Akgül and Karaküçük, was used to measure participants’ attitudes and tendencies towards leisure time activities. The scale is a valid and reliable tool for measuring how individuals evaluate their leisure time, which activities they are oriented towards, and their motivation to participate in leisure activities. Consisting of 30 items, the scale consists of three sub-dimensions: cognitive, affective, and behavioral. The Cronbach’s alpha (α) reliability coefficient of the scale was found to be 0.89 in general and 0.85 in the cognitive dimension, 0.83 in the affective dimension, and 0.87 in the behavioral dimension, respectively [16].

#### 2.2.3. Physical Activity Enjoyment Scale

The Physical Activity Enjoyment Scale, developed by Özkurt et al., was used to determine the enjoyment levels of the participants during participation in physical activities. The scale is a valid and reliable tool for measuring satisfaction and enjoyment that individuals feel during physical activity. The scale consists of multidimensional items that reflect the participants’ emotional reactions to physical activity. The data obtained in the study were calculated as a high reliability coefficient of the scale (Cronbach’s alpha = 0.92) [17].

#### 2.2.4. Digital Game Addiction Scale

In this study, the Digital Game Addiction Scale developed by Irmak and Erdoğan was used to determine participants’ digital game addiction levels. The scale is a valid and reliable tool for measuring individuals’ digital game-playing habits, game-playing time, and addiction levels. The Cronbach’s alpha (α) reliability coefficient of the scale, which consists of one dimension and contains seven items, was calculated as 0.80 [18].

### 2.3. Machine Learning Approaches

In this study, different hybrid ML algorithms and routine regression analysis results were compared with the GBM + RF hybrid model to predict SA and GA variables. LTM and EPA independent variables were used to predict these outcomes. Before the analyses, missing data were identified and normalized. To improve the model performance, the independent variables were scaled using the StandardScaler. Subsequently, all analyses were compared, and the model with the best performance was selected [19]. All models were tested using 5-fold cross-validation.(1)$${CV}_{score}=\frac{1}{k}{\left(x+a\right)}^{n}=\sum_{i=1}^{k}R_{i}^{2}$$

The performances of all models for ML algorithms were analyzed with the R^2^, mean absolute error (MAE), and mean square error (MSE) metrics.(2)$$R^{2}=1-\frac{{SS}_{residual}}{{SS}_{total}}$$(3)$$MAE=\frac{1}{n}\sum_{i=1}^{n}\left|y_{i}-{\hat{y}}_{i}\right|$$(4)$$MSE=\frac{1}{n}\sum_{i=1}^{n}{\left(y_{i}-{\hat{y}}_{i}\right)}^{2}$$

#### Proposed Hybrid GBM + RF

In this study, various ML algorithms and the GBM + RF algorithm were used to compare the model performance in data analysis and prediction tasks. The methods evaluated included artificial neural networks (ANNs), gradient boosting, logistic regression, decision trees, and random forest. These algorithms have been tested both individually and, in some cases, as part of hybrid models combining multiple techniques [20,21,22,23]. The hybrid modeling approach aimed to enhance generalizability and reduce error rates by leveraging the strengths of various algorithms. Because each method performs differently, depending on the structure and features of the data, this study evaluated the diversity of approaches and compared their performance to identify the most effective methods.

Random forest (RF) is an ensemble learning method in which multiple decision trees work together to produce strong predictive results [24]. It is widely used for both classification and regression problems because of its high accuracy, low variance, and strong generalization capability. The algorithm is based on the principle of bagging, where numerous independent decision trees are built using randomly selected bootstrap samples from the training data. Each tree is trained on a random subset of the data and a random subset of features, which increases model diversity and helps reduce the risk of overfitting. The prediction process is straightforward: each tree makes an independent prediction, and these predictions are aggregated to produce a final output. For regression problems, the final prediction is calculated by taking the arithmetic mean of the outputs of all trees.(5)$$\hat{y}\left(x\right)=\frac{1}{M}\sum_{m=1}^{M}T_{m}(x)$$

Here, the following applies:$\hat{y}\left(x\right)\;$
represents the final prediction made by the model for observation *x*;$T_{m}(x)$
denotes the prediction made by the m-th decision tree for the same observation;and M indicates the total number of trees in the model.

RF models are particularly effective when working with high-dimensional data sets, missing values, or noisy data. They also demonstrate a strong tolerance for complex inter-variable relationships. One of the key advantages of this method is its interpretability; thanks to feature importance scores, it is possible to assess how much each variable contributes to the overall predictive performance of the model.

A gradient-boosting machine (GBM) is a powerful prediction algorithm that falls under the category of ensemble learning methods. It works by sequentially training weak learners, typically decision trees, where each new model focuses on correcting the errors made via the previous model. The GBM can deliver high predictive accuracy, especially in data sets with nonlinear and complex relationships [25].

The GBM algorithm used in this study is based on a sequential learning strategy that minimizes prediction errors iteratively. The model construction process can be described in three main mathematical steps: initial prediction, error estimation based on the negative gradient, and model updating. These steps are detailed below along with their respective equations. Initial prediction (Equation (6)): The first step involves determining a constant initial value for the entire data set. This value is typically a constant score that minimizes the chosen loss function.(6)$$F_{0}\left(x\right)=arg\;\underset{c}{\min}\sum_{i=1}^{n}L(y_{i},c)$$

Here, $L\left(y_{i},c\right)$ represents the loss between the true value, *y*ᵢ, and the constant prediction, c. This forms the base prediction for the zeroth iteration of the model.

Error estimation—negative gradient: In the second step, the negative gradient of the loss function with respect to the predictions is calculated for each observation. These residuals serve as target values for training the next weak learner.(7)$$r_{i}^{\left(m\right)}=-{\left[\frac{\partial L(y_{i},F\left(x_{i}\right))}{\partial F\left(x_{i}\right)}\right]}_{F=F_{m}-1}$$

Here, $r_{i}^{\left(m\right)}$ denotes the negative gradient computed for the i−th observation at the m−th iteration.

Model updating (Equation (8)): Finally, the model output is updated by adding the output of the new learner to the previous model prediction, scaled by a learning rate η.(8)$$F_{m}\left(x\right)=F_{m-1}\left(x\right)+\eta .h_{m}(x)$$

where the following applies:
$F_{m}\left(x\right)$ is the updated prediction at iteration m;$h_{m}\left(x\right)$ is the output from the new learner;η is the learning rate.

These three steps are repeated iteratively. At each step, the model reduces the prediction errors, gradually improving its accuracy. This sequential structure enables the GBM to perform exceptionally well, particularly on data sets with nonlinear and complex relationships.

In this study, a hybrid model was developed by combining two tree-based ensemble learning algorithms: GBM and RF. These two algorithms complement each other in terms of their learning approaches. GBM builds new trees that focus on correcting the errors made by previous predictions, thereby incrementally improving the model’s accuracy. This sequential optimization process allows the GBM to perform exceptionally well, especially in data sets with complex and nonlinear relationships.

In contrast, RF uses a bagging approach, combining the results of numerous decision trees trained independently on random subsets of the data. This structure enhances the generalization ability of the model and provides strong resistance to overfitting. RF is also known to produce stable predictions, even with high-dimensional and noisy data.

In the first stage, GBM learned the complex and non-linear relationships between structural variables and the target variable; then, GBM outputs (probability scores) were fed into the RF model in the second stage to produce final predictions. This structure combines GBM’s error reduction capability with RF’s variance-reducing effect, resulting in superior performance in terms of both accuracy and generalizability. In the proposed hybrid model, GBM was first used to learn the relationships between structural variables and the target variable. The output scores generated via GBM were then used as inputs for the RF model to produce the final predictions. By combining GBM’s strength in error reduction with RF’s power in reducing variance, the resulting model achieved higher accuracy and better generalizability. This approach offers an effective solution for challenges like high dimensionality, multicollinearity, and data scarcity, which are common in the prediction of biological activity in chemical compounds. In this context, the ML process applied in our study is presented in detail in Figure 1.

The whole process of the proposed hybrid GBM + RF model is expressed as a single function as in Equation (9):(9)$$\hat{y}\left(x\right)=\frac{1}{K}\sum_{k=1}^{K}T_{k}(F_{M}(x))$$

In the second stage of the study, ML techniques were employed to estimate the theoretical biological activity of a set of pyrazole derivatives. To reduce the dimensionality of the data set and focus on the most informative molecular descriptors, feature selection was first carried out using recursive feature elimination (RFE). In some cases, k-fold cross-validation was also utilized to better assess model stability and generalizability. A range of algorithms were tested, including logistic regression, decision trees, RF, gradient boosting, and artificial neural networks. In addition to these individual models, hybrid approaches were explored to leverage complementary strengths. Notably, the RF + ANN combination and the custom-designed GBM + RF hybrid model delivered promising results. Model evaluation was based not only on standard performance metrics such as R^2^, MAE, and MSE but also on the Q^2^ statistic to ensure predictive reliability on unseen data. Overall, the proposed GBM + RF model stood out for its accuracy and robustness.

### 2.4. Statistical Analyses

In this study, statistical analyses were performed with Python 3.13.2. In order to evaluate the SA and DA levels of the participants, a number of analyses were performed in addition to ML analyses. In this process, normality analysis, descriptive statistics, a comparison of SA, GA, LTM, and EPA scores between genders, and multiple regression analysis were performed. The normality of the data was analyzed using the Kolmogorov-Smirnov test, and skewness and kurtosis values were also examined. After this examination, it was determined that the dependent (SA, GA) and independent variables (LTM, EPA) were normally distributed. In addition, the Omnibus test and the Jarque–Bera test were applied to determine the normality of the data set. An independent-sample *t*-test was applied to evaluate the scores of the participants according to gender. In addition, multiple regression analysis was applied to determine the relationship between SA and GA scores and LTM and EPA variables. The performance level of the regression models was interpreted by looking at the R^2^, adjusted R^2^, AIC, BIC, and Durbin–Watson criteria. In order to increase the reliability of the results of the significance level in the analyses, 95% confidence interval (CI) scores were also calculated. Cohen’s d effect sizes were also calculated to evaluate the magnitude of the difference between the groups. Accordingly, an effect size of less than 0.2 indicates a small effect, values between 0.2 and 0.5 represent a moderate effect, and those greater than 0.8 correspond to a large effect. The statistical significance level for this study was set to 0.05 (*p* < 0.05).

## 3. Results

This section presents descriptive statistics on participants’ levels of smartphone and digital game addiction, as well as their attitudes toward leisure time management and physical activity.

According to Table 1, the SA (27.16 ± 7.90), GA (15.67 ± 6.72), LTM (47.27 ± 7.82), and EPA (41.54 ± 13.00) were found.

Table 2 shows the results of the least squares (OLS) regression analysis of the participants’ SA, LTM, and EPA results. According to the results, the model accounts for a low proportion of the dependent variable (SA) (R^2^ = 0.015). Therefore, the explanatory power of the model is also low (Adj R^2^ = 0.013). However, a significant difference was found between the independent variables affecting SA in this model [F(2, 1107) = 8.367, *p* < *0*.001)]. When the results of the regression analysis between the independent variables and SA were analyzed, as the LTM score increased, there was a significant decrease in the SA value (coef = −0.879, std err = 0.031, t = −2.811, *p* = 0.005). As the EPA score increases, there is a significant decrease in the SA value (coef = −0.0402, std err = 0.019, t = −2.135, *p* = 0.033).

In Figure 2, the 3D regression analysis illustrates the relationship between SA and the independent variables LTM and EPA. Each blue point represents a participant and is plotted in a three-dimensional space according to the total reported free time (X-axis), enjoyment of physical activity (Y-axis), and the corresponding SA predicted by the model (Z-axis). The presence of high-density points at the center of the three axes indicates a central trend in user behavior. The spatial distribution of data points suggests that free time behaviors have non-linear and potentially interactive effects on predicted smartphone addiction. The model visualization supports the hypothesis that increased enjoyment from physical activity and structured use of free time may be inversely related to smartphone addiction tendencies.

Table 3 shows the results of the least squares (OLS) regression analysis of the participants’ GA, LTM, and EPA results. According to the results, the model accounts for a low proportion of the dependent variable (GA) (R^2^ = 0.013). Therefore, the explanatory power of the model is also low (Adj R^2^= 0.011). However, a significant difference was found between the independent variables affecting GA in this model [F(2, 1107) = 7.353, *p* < *0*.001)]. When the results of the regression analysis between the independent variables and GA were analyzed, as the LTM score increased, there was a significant decrease in the GA value (coef = −0.1018, std err = 0.027, t = −3.824, *p* < 0.001). No relationship was found between EPA and GA (*p* > 0.05).

In Figure 3, the 3D regression analysis illustrates the relationship between SA, LTM, and EPA. Each data point represents an individual participant (N ≈ X) plotted according to their reported total free time (X-axis), their enjoyment of physical activity (Y-axis), and the ML model’s predicted digital game addiction score (Z-axis). A dense central clustering indicates that most predictions are concentrated around the midpoints of both the predictors and the outcomes. However, the spatial distribution indicates non-linear interactions between predictors and the target variable. This pattern supports the hypothesis that EPA and the structured use of free time are inversely related to predicted levels of GA.

Table 4 presents the performance metrics (MAE, MSE, and R^2^) of ML models developed to predict participants’ SA based on their LTM and EPA scores. The evaluated models include linear algorithms (lasso, ElasticNet), tree-based ensemble methods (random forest, gradient boosting, AdaBoost, bagging), support vector and nearest neighbor methods (SVR, KNN), artificial neural networks (MLP), an advanced boosting model (LightGBM), and a hybrid approach (GBM + RF). Among all models, the Hybrid GBM + RF model achieved the highest predictive performance with the lowest MAE (0.095), MSE (0.010), and a strong R^2^ value (0.9299), indicating its robustness in capturing the relationship between SA, LTM, and EPA. Among linear models, lasso regression (MAE = 0.105, MSE = 0.011, R^2^ = 0.9296) and ElasticNet (MAE = 0.112, MSE = 0.012, R^2^ = 0.9296) yielded similarly high performance with low error rates. Among the tree-based ensemble methods, gradient boosting (MAE = 0.195, R^2^ = 0.9260), random forest (MAE = 0.210, R^2^ = 0.9249), bagging (MAE = 0.208, R^2^ = 0.9250), and MLP (MAE = 0.200, R^2^ = 0.9259) demonstrated competitive performance, though with slightly higher error rates compared to linear models. LightGBM also performed strongly (MAE = 0.135, MSE = 0.018, R^2^ = 0.9288), reinforcing its utility for this prediction task. In contrast, models such as AdaBoost (MAE = 0.300, R^2^ = 0.8693), SVR (MAE = 0.310, R^2^ = 0.8615), and KNN regression (MAE = 0.295, R^2^ = 0.8729) showed relatively lower predictive accuracy. These results underscore the effectiveness of ensemble and regularized linear models in modeling cognitive performance metrics, while highlighting the comparatively limited utility of simpler or non-ensemble models for this data set.

Table 5 presents the performance results of ML models used to predict participants’ GA based on their LTM, EPA scores. Similar to Table 4, the evaluated models include linear regressions (lasso, ElasticNet), ensemble techniques (random forest, gradient boosting, AdaBoost, bagging), support vector and neighbor-based models (SVR, KNN), multilayer perceptron (MLP), LightGBM, and a hybrid approach combining GBM and random forest. Among all models, the Hybrid GBM + RF model again delivered the best performance (MAE = 0.090, MSE = 0.014, R^2^ = 0.9699), indicating high predictive accuracy and minimal error. Among the linear models, lasso regression (MAE = 0.165, MSE = 0.027, R^2^ = 0.9693) showed strong performance, closely followed by ElasticNet regression (MAE = 0.178, MSE = 0.032, R^2^ = 0.9681). Similarly, LightGBM (MAE = 0.165, MSE = 0.029, R^2^ = 0.9691) and MLP (MAE = 0.272, MSE = 0.074, R^2^ = 0.9689) also exhibited high levels of accuracy. Ensemble-based models like random forest (MAE = 0.300, MSE = 0.090, R^2^ = 0.9667), gradient boosting (MAE = 0.293, MSE = 0.085, R^2^ = 0.9675), and bagging (MAE = 0.315, MSE = 0.093, R^2^ = 0.9674) demonstrated consistent results with moderate error values and strong R^2^ scores around 0.967. In contrast, AdaBoost (MAE = 0.380, MSE = 0.145, R^2^ = 0.8995), SVR (MAE = 0.400, MSE = 0.160, R^2^ = 0.9036), and KNN regression (MAE = 0.395, MSE = 0.156, R^2^ = 0.8913) exhibited relatively lower performance, with KNN showing the lowest R^2^ and among the highest error metrics.

## 4. Discussion

This study was conducted to investigate the relationship between SA and GA levels and LTM and EPA scores among adolescents using classical regression analyses and various ML algorithms. In the linear regression analysis, it was found that the LTM and EPA variables showed a negative correlation with SA. On the other hand, while LTM had a negative effect on GA, no significant effect of EPA on GA was observed. Although both dependent variables were found to be influenced by LTM and EPA, the explanatory power level (R^2^) of the regression models created was determined to be quite low. In the analyses where ML algorithms were tested, the highest performance in SA and GA prediction was achieved with the Hybrid GBM + RF model. In particular, the Hybrid GBM + RF model stood out with the lowest error rates and highest R^2^ values in both outputs. ML algorithms demonstrated higher explanatory power compared to classical linear regression analysis. In line with these findings, our research hypothesis was confirmed. To the best of our knowledge, this study is the first to compare SA and GA levels using LTM and EPA variables with both classical regression and ML methods, and to utilize the GBM + RF hybrid model in predicting psychological disorders.

In other studies, examining the relationship between LTM and participation in physical activity and SA, there was a negative correlation between both independent samples and SA [11,26]. Abbasi et al. argued that the main causes of SA are entertainment, social networking sites, and game-related uses. He also emphasized that physical activity significantly reduces the level of SA and minimizes the negative effects of SA on academic achievement [27]. In the study conducted by Al-Amri et al., the effects of SA on cognitive function and physical activity were examined, and it was pointed out that SA levels were low in physically active individuals, as well as the usual low level of physical activity in individuals with addiction [28]. When analyzed in this context, the results of our study and the studies in the literature show similar results. In these studies, the independent variables of non-dependent and dependent individuals were compared, whereas in our study, a regression model was established between LTM and EPA and SA, and a highly significant interaction was found. Although the effects of independent variables different from the variables we analyzed in our study on SA have been examined, Aker et al. concluded that depression, anxiety, insomnia, and family social support predicted SA in university students [29]. Studies also suggest that gender is an important factor in the SA and GA levels of participants [11,26]. Abbasi et al. suggested that the main causes of SA are entertainment, social networking sites, and gaming. Additionally, they emphasized that physical activity significantly reduces SA levels and minimizes the negative effects of SA on academic performance [27]. In a study conducted by Al-Amri et al., the effects of SA on cognitive functions and physical activity were examined, and it was noted that SA levels were low in physically active individuals, while physical activity levels were generally low in individuals with addiction [28]. Aker et al. concluded that depression, anxiety, insomnia, and family social support predicted SA among university students [29]. In our study, a separate algorithm was not tested for gender differences. As understood from the literature, the purpose of phone use may differ between genders [30]; therefore, in future studies with larger sample sizes, analyses should be conducted while considering this factor.

GA has emerged as a global issue, particularly among young individuals, with profound effects on social relationships, academic performance, and mental health. Addressing GA requires preventive strategies, and research has highlighted the critical role of behaviors such as LTM, self-control, and self-efficacy in mitigating its effects [31]. Simultaneously, the potential for the occurrence of a behavioral disorder, such as GA, in those who are physically active is quite low [32]. The 3D modeling analysis conducted in this study identified LTM as a significant predictor of GA, with results indicating a negative association between both LTM and EPA (enjoyment of physical activity) with GA levels. This suggests that adolescents who effectively manage their leisure time and engage in physical activity are less likely to develop GA. These results align with previous research demonstrating that individuals with high self-control exhibit lower GA [33]. Considering that individuals with high self-control plan their leisure time positively, it is thought that LTM is among the important components of the positive decrease in GA.

In recent years, ML methods have been frequently used to predict SA and GA. Some studies have focused on predicting risky groups that may become addicted to smartphones in the future as a result of a certain period of follow-up [34]. Some studies have focused on how the level of SA in parents affects the situation of offspring [35]. Others have focused on the effects of different independent variables on SA [36]. Similarly, studies have been conducted to predict GA with different independent variables, and the effect of different independent variables has been examined [37]. In this context, ML’s ability to process large data sets and detect complex patterns of addiction with high accuracy makes it an important method for analyzing addiction types that are affected by many factors, such as SA and GA, which have been increasing in recent years.

In this study, ML algorithms and classical regression models were used to compare SA and GA levels based on the LTM and EPA variables. The results indicate that the highest prediction performance for SA was achieved using the Hybrid GBM + RF model (R^2^ = 0.9299), and the same model also provided the best results for GA prediction (R^2^ = 0.9699). In contrast, classical linear regression models exhibited significantly lower explanatory power (e.g., R^2^ = 0.015 for SA and R^2^ = 0.013 for GA) [35], highlighting a significant difference between the two approaches and emphasizing the methodological importance of this distinction. This inconsistency can be attributed to the ability of ML algorithms to effectively capture complex, non-linear, and multi-variable relationships that traditional regression techniques often fail to adequately model. Given the limited number of independent variables (i.e., LTM and EPA), the reported accuracy may reflect optimization within a narrowly defined feature space, rather than a robust generalization ability. Therefore, we emphasize the importance of evaluating model performance not only based on R^2^ values but also in conjunction with MAE and MSE metrics to provide a more comprehensive and accurate assessment. In conclusion, while classical linear models may be insufficient in explaining the complexity of behavioral phenomena, ML algorithms can also yield misleadingly optimistic results if not applied and interpreted carefully. The results of our study were similar to those obtained using a hybrid model in previous machine learning studies. In their study, Gali et al. tested a hybrid SVM model to predict decision-making abilities and achieved higher accuracy [38]. In a model similar to ours, where Ding et al. used a hybrid model to determine participants’ stress levels, participants’ stress levels were predicted correctly at a rate of 100%. This study also faced similar limitations. In this study, the data set consisted solely of data based on sleep and physiological sensor measurements. The study noted that the failure to fully incorporate cognitive and environmental determinants of stress into the model limited the generalizability of the findings [39]. To enhance both theoretical validity and practical applicability, future research should incorporate a broader range of cognitive, emotional, and contextual determinants. This will support the development of models that are both more generalizable and better represent real-world behavioral dynamics.

While this study provides valuable insights into SA and GA in relation to LTM and EPA, certain limitations should be acknowledged. First, only LTM and EPA scores were analyzed as independent variables, which may have limited the model’s predictive power. Future studies could enhance model performance by incorporating additional factors such as socioeconomic status, region of residence, educational background, and environmental influences. Including these variables could provide a more comprehensive understanding of the factors contributing to SA and GA. Second, this study focused exclusively on adolescents, which may limit the generalizability of the findings to other age groups. Future research should consider broader age ranges to allow comparisons across different developmental stages and improve the applicability of the results. Another limitation is the self-reported nature of the data. As participants completed the questionnaires themselves, there was a potential for response bias, social desirability effects, or reluctance to answer certain questions. Implementing objective measures or incorporating parent and teacher evaluations could enhance the reliability of the findings. Despite these limitations, this study provides important initial evidence regarding the relationship between digital addiction, leisure time management, and physical activity. Future research should build upon these findings by expanding the variables analyzed, diversifying the sample population, and utilizing more objective assessment methods. One important limitation is that separate analyses were not conducted according to gender. In future studies, it may be recommended that gender be added to the model as a variable.

## 5. Conclusions

In our study, the relationships between SA and GA levels and the LTM and EPA variables were examined using multiple linear regression analyses and various ML algorithms. The regression analysis results revealed that SA levels were significantly and negatively affected by both LTM and EPA, whereas GA was negatively associated only with LTM. These findings suggest that as adolescents’ capacity for LTM use and effective performance analysis increases, their levels of addiction (i.e., lower SA and GA scores) tend to decrease. However, the explanatory power (R^2^) of the regression models was relatively limited. In contrast, analyses performed using ML algorithms yielded considerably higher predictive performances. Specifically, the Hybrid GBM + RF model demonstrated the best results in estimating both SA and GA levels (R^2^ = 0.9299 for SA; R^2^ = 0.9699 for GA), highlighting the capacity of ML methods to better model complex and nonlinear relationships than traditional approaches. Despite the high R^2^ values observed in these models, the overall explanatory power remains limited compared to the existing literature, which may be attributed to the inclusion of only two independent variables (LTM and EPA) in the model. To enhance predictive accuracy and model robustness in future research, it is recommended that additional individual and environmental variables, such as screen time, age, gender, social environment, anxiety levels, and family interaction, be incorporated. Moreover, promoting educational and social activities that encourage adolescents to spend their free time constructively may help reduce addiction tendencies. In particular, programs that support participation in physical activities can play a vital role in the prevention of digital addictions, including SA and IA (internet addiction). Finally, parental involvement in structuring screen time and fostering self-control and time management skills is essential for supporting healthier digital habits among adolescents. Future studies should incorporate additional variables, such as socioeconomic status, screen time, digital media habits, and psychological well-being, to enhance model accuracy and generalizability. Expanding the sample to include different age groups and employing mixed-method approaches could also provide deeper insights into the underlying factors of digital addiction.

## Figures and Tables

**Figure 1 healthcare-13-01805-f001:**
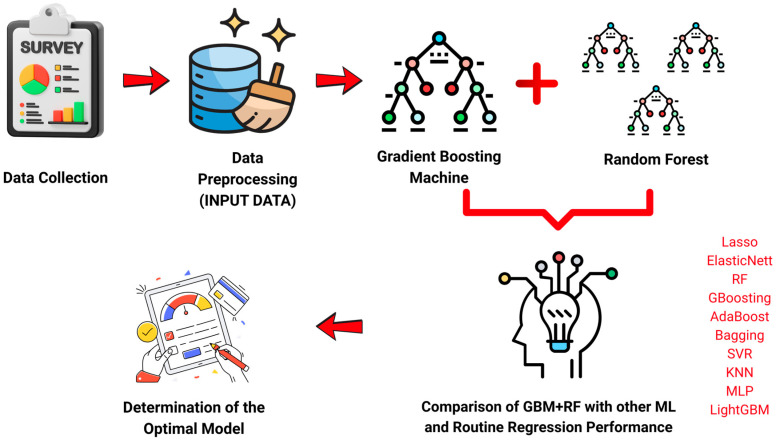
ML approaches: input data = LTM and EPA.

**Figure 2 healthcare-13-01805-f002:**
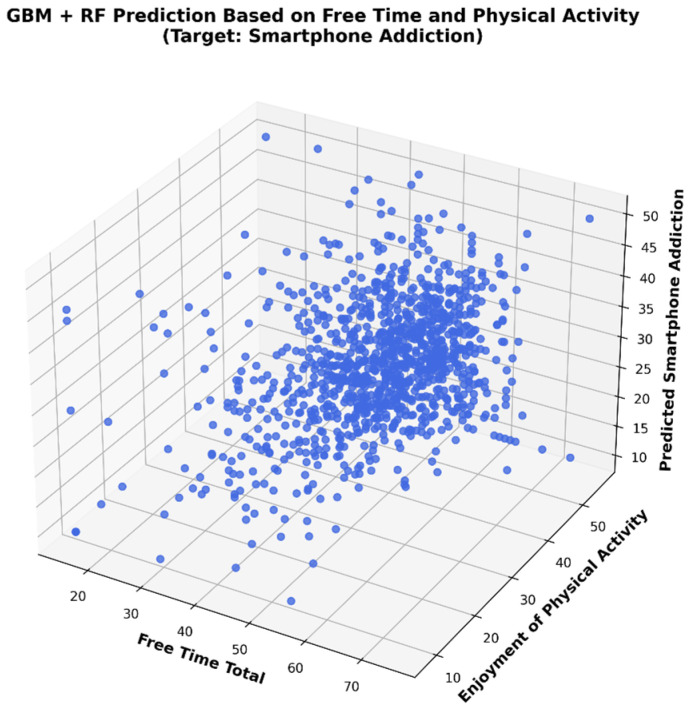
Illustrates 3D analysis of regression analysis between SA and independent variables.

**Figure 3 healthcare-13-01805-f003:**
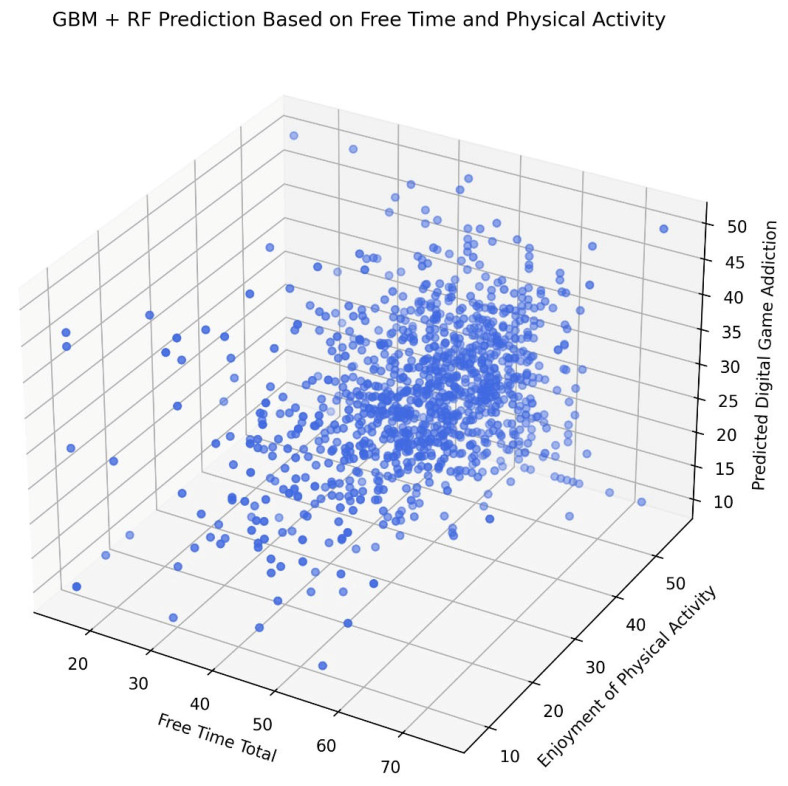
Three-dimensional analysis of regression analysis between GA and independent variables.

**Table 1 healthcare-13-01805-t001:** Baseline information of SA, GA, LTM, and EPA scores.

Parameters	N	M	SD	Minimum	Maximum
SA (score)	1107	27.16	7.90	10.0	50.0
GA (score)		15.67	6.72	7.0	35.0
LTM (score)		47.27	7.82	15.0	75.0
EPA (score)		41.54	13.00	8.0	56.0

SA: Smartphone addiction; GA: digital game addiction; LTM: leisure time management; EPA: enjoyment of physical activity.

**Table 2 healthcare-13-01805-t002:** Results of regression analysis between SA, EPA, and LTM.

Parameters	Coef	Std Err	t	*p*-Value	%95 CI	
LTM	−0.0879	0.031	−2.811	0.005	−0.149	−0.027
EPA	−0.0402	0.019	−2.135	0.033	−0.077	−0.003
Result of the Regression Model						
R^2^	0.015					
Adj. R^2^	0.013					
F	8.367					
*p*-value	<0.001					
AIC	7708					
BIC	7723					
Omnibus	6.037					
Durbin–Watson	1.950					

Abbreviation: LTM: leisure time management; EPA: enjoyment of physical activity; AIC: Akaike Information Criterion; BIC: Bayesian Information Criterion.

**Table 3 healthcare-13-01805-t003:** Results of regression analysis between GA, EPA, and LTM.

Parameters	Coef	Std Err	t	*p*-Value	%95 CI	
LTM	−0.1018	0.027	−3.824	<0.001	−0.154	−0.50
EPA	0.0205	0.016	1.277	0.202	−0.011	0.052
Result of the Regression Model						
R^2^	0.013					
Adj. R^2^	0.011					
F	7.353					
*p*-value	<0.001					
AIC	7351					
BIC	7366					
Omnibus	61.741					
Durbin–Watson	1.930					

Abbreviation: LTM: leisure time management; EPA: enjoyment of physical activity; AIC: Akaike Information Criterion; BIC: Bayesian Information Criterion.

**Table 4 healthcare-13-01805-t004:** Performance results of machine learning models between SA, LTM, and EPA.

Model	MAE	MSE	R^2^
Lasso Regression	0.105	0.011	0.9296
ElasticNet	0.112	0.012	0.9296
Random Forest	0.210	0.045	0.9249
Gradient Boosting	0.195	0.039	0.9260
AdaBoost	0.300	0.090	0.8693
Bagging	0.208	0.044	0.9250
SVR	0.310	0.096	0.8615
KNN Regression	0.295	0.085	0.8729
MLP (Neural Net)	0.200	0.041	0.9259
LightGBM	0.135	0.018	0.9288
Hybrid GBM + RF	0.095	0.010	0.9299

**Table 5 healthcare-13-01805-t005:** Performance results of machine learning models between GA, LTM, and EPA.

Model	MAE	MSE	R^2^
Lasso Regression	0.165	0.027	0.9693
ElasticNet Regression	0.178	0.032	0.9681
Random Forest	0.300	0.090	0.9667
Gradient Boosting	0.293	0.085	0.9675
AdaBoost	0.380	0.145	0.8995
Bagging	0.315	0.093	0.9674
SVR	0.400	0.160	0.9036
KNN Regression	0.395	0.156	0.8913
MLP (Neural Network)	0.272	0.074	0.9689
LightGBM	0.165	0.029	0.9691
Hybrid GBM + RF	0.090	0.014	0.9699

## Data Availability

The original contributions presented in this study are included in the article. Further inquiries can be directed to the corresponding author.

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
