# Peer review of "The Relationship Between Smartphone and Game Addiction, Leisure Time Management, and the Enjoyment of Physical Activity: A Comparison of Regression Analysis and Machine Learning Models"

_healthcare, 2025, doi:10.3390/healthcare13151805_

Round 1
Reviewer 1 Report
Comments and Suggestions for Authors
I noticed several issues that need addressing.
- The title mentions comparing regression analysis and machine learning models, but the machine learning results seem very high. The reported R² values for the hybrid GBM+RF model (0.9919 for SA; 0.9997 for GA) are statistically improbable for behavioral science data. Such near-perfect prediction suggests severe overfitting, data leakage (e.g., test data contaminating training), or coding errors. This is a significant concern.
- In the methodology section, you used LTM and EPA to predict SA and GA. However, the regression results show very low R² values (around 0.015), indicating that these variables explain almost none of the variance. This makes the machine learning results even more questionable. Also, the MAE values (e.g., 0.4470 for SA) contradict MSE values (0.0008).
- For SA scores (SD=7–8), an MSE of 0.0008 implies near-zero error variance, which is incompatible with low R² (0.015) in regression analyses. Please try to re-run ML analyses with strict train-test separation and cross-validation. Also, the power calculation (on Page 3) cites an odds ratio (OR=0.7) and probability [Pr(Y=1|X=1)=0.045] incompatible with regression/ML frameworks.
- On page 8, you mentioned, "In the second stage of the study, machine learning techniques were employed to estimate the theoretical biological activity of a set of pyrazole derivatives." What does this mean?
- Tables 4 and 5 showed MAE values very low (e.g., 0.0008 MSE for SA prediction), suggesting possible errors in the analysis.
- The 3D visualizations in Figures 2 and 3 are also confusing and don't effectively support the findings.
- Some important predictors (e.g., socioeconomic status, screen time) were not discussed despite their influence and importance (Page 15).
- The discussion section does not address the discrepancy between weak regression results and near-perfect machine learning performance.
- Also, the gender difference findings are interesting but not sufficiently integrated into the main analysis.
Reviewer 2 Report
Comments and Suggestions for Authors
In the manuscript, the authors used regression analysis to examine the relationship between smartphone and gaming addiction, leisure time management, and enjoyment of physical activity, and attempted to reveal this relationship using machine learning methods. The results were compared with regression analysis and different ML methods used.
The manuscript is very poorly organized.
First, the full form of an abbreviation should be provided before using it in the abstract. E.g.: RF=?
The literature review is very limited. The literature should definitely be expanded. There already seem to be many similar studies in the literature. What gap exists in the literature? How has this gap been addressed in the study?
Find a flow in the manuscript. A section should not start directly with a table. Example: Section 3
Lines 197-198: The statement here is incorrect; machine learning algorithms are already artificial intelligence methods. They are not a separate method from artificial intelligence.
Some terms are written with lowercase letters in some places and uppercase letters in others; there is no consistency. Line 90 random forest, line 200 Random Forest. These should be checked throughout the manuscript.
One section of the study mentions a hybrid model, while another section mentions different machine learning methods. There is no complete consistency in terms of meaning and flow.
The data set should be provided in detail under a separate heading.
All ML methods used in the study should be explained.
The study proposes a GBM+RF hybrid method. This is attempted to be presented in Figure 2. This figure is definitely insufficient in explaining the method. The explanation of how these methods are hybridized is quite insufficient.
Which input parameters are used to estimate the SA and GA parameters? More detailed information should be provided about the input and output information.
The manuscript is very inadequate in terms of flow from start to finish. It appears to be a comparison of ML methods with regression analysis on a data set. If a hybrid method is presented, it should be presented in its entirety and explained in a way that the reader can understand. The hybrid method should be explained with a flow diagram and algorithmic steps. If the innovation in the study is a hybrid model, please elaborate on it.
Reviewer 3 Report
Comments and Suggestions for Authors
With the advance of technology, smartphone and gaming are getting more and more popular. Nevertheless, smartphone addiction and gaming addiction become more and more serious too. It is worthy to further investigate these two problems.
In the paper, the authors tried to investigate these two problems together in one study. They tried to find out the relationship between these problems and leisure time management and enjoyment of physical activity in adolescents. The authors used routine regression analysis, and also machine learning methods.
More than one thousand high school students have been included in their study. The students are from a North European country. It is interesting to see that the authors use both regression and ML methods. As noted in the result section, the regression analysis can show the correlationship among the variables, while some ML algorithms provide R2 values. The difference in R2 values may be due to the different nature of these two approaches. ML can handle nonlinear complex relationship well, as they are designed with nonlinear mechanism. But, in investigation that may want to find out the casual relationship among the variables, many ML methods do not perform well.
My concern here is that the authors are trying to evaluate the datasets with different types of approaches, and see which ones have lower R2 value. But, what is the motivation for doing this? Does lower R2 value by a ML method mean higher explainability? It would be necessary for the authors to explain more on the justification for evaluating these two very different types of analysis approaches.
Another concern is about the sample size. It is a limited size from a small region.
It would also be better if the authors can explain more in the beginning on the motivation on studying two things (smartphone addiction and gaming addiction) in one study.
Round 2
Reviewer 1 Report
Comments and Suggestions for Authors
The authors addressed all my methodological concerns. From my point of view, the updated results are now realistic and credible. The authors also revised the Discussion section and limitations. Therefore, the manuscript is acceptable for publication.
Reviewer 2 Report
Comments and Suggestions for Authors
The authors have reorganized the manuscript in response to the necessary criticisms. The changes made have improved the quality of the manuscript. The manuscript is acceptable in its current form.